# Sedentary Behaviour and Its Correlates Among Older Adults in Malaysia

**DOI:** 10.3390/healthcare13020160

**Published:** 2025-01-15

**Authors:** Chee Cheong Kee, Lay Kim Tan, Yong Kang Cheah, Chien Huey Teh, Hui Li Lim, Yoon Ling Cheong, Mohd Azahadi Omar, Sumarni Mohd Ghazali

**Affiliations:** 1Sector for Biostatistics and Data Repository, National Institutes of Health, Ministry of Health Malaysia, Shah Alam 40170, Malaysia; tanlk@moh.gov.my (L.K.T.); drazahadi@moh.gov.my (M.A.O.); 2School of Economics, Finance and Banking, College of Business, Universiti Utara Malaysia, Sintok 06010, Malaysia; yong@uum.edu.my; 3Institute for Medical Research, National Institutes of Health, Ministry of Health Malaysia, Setia Alam 40170, Malaysia; chienhuey@moh.gov.my (C.H.T.); cheongyl@moh.gov.my (Y.L.C.); sumarni.mg@moh.gov.my (S.M.G.); 4Institute for Clinical Research, National Institutes of Health, Ministry of Health Malaysia, Setia Alam 40170, Malaysia; lim.hl@moh.gov.my

**Keywords:** sedentary behaviors, older persons, correlates, aging, older adults

## Abstract

**Background:** Sedentary behaviors (SBs), which are low-energy, wakeful activities performed in a sitting, reclining, or lying posture, are independently associated with numerous adverse health outcomes, including mental health disorders, non-communicable diseases, and an increased risk of mortality. This study investigated associations between sociodemographic characteristics, lifestyle factors, mental health, nutritional status, social support, functional limitations, and SB among older persons in Malaysia. **Methods:** Data from 3977 individuals aged 60 years and above, extracted from the National Health and Morbidity Survey (NHMS) 2018, were analyzed using complex samples logistic regression. **Results:** The prevalence of sedentary behavior, defined as sitting or reclining for 8 or more hours per day, among the surveyed population was 23.2%. Older age (≥75 years) was significantly associated with higher odds of SB (AORs 1.58 to 2.76, *p* < 0.001 to *p* = 0.001). Unemployment (AOR = 1.32, *p* = 0.020) and indigenous Sabah and Sarawak ethnicity (AOR = 2.48, *p* = 0.007) were also linked to increased odds of SB. Conversely, individuals with a monthly income of MYR 1000-1999 had lower odds of SB compared to those earning ≥MYR 2000 (AOR = 0.64, *p* = 0.022), and those at risk of malnutrition were also less likely to engage in SB (AOR = 0.68, *p* = 0.031). No significant associations were found between SB and sex, marital status, educational level, or chronic illness. **Conclusions:** These findings suggest that public health initiatives to reduce SB among older adults should prioritize the oldest aged, unemployed, and specific ethnic communities, as well as addressing nutritional risk to promote healthier aging among older persons in Malaysia.

## 1. Introduction

In 2020, the population in Malaysia aged 65 and above was estimated at 2.32 million, or 7.0% of the total population of 33.2 million. Based on the current trajectory, by 2050, the population aged 65 and above is estimated to reach 7.14 million, or 17.4% of the population, making Malaysia an aged nation [1]. Population aging has significant implications for the healthcare system, as more social and financial support will be needed to meet the increasing demand for healthcare services. This is due to the growing prevalence of non-communicable diseases (NCDs) such as cardiovascular diseases, diabetes, hypertension, dementia, and cancer among the older population. Therefore, supporting healthy aging is crucial, with an emphasis on identifying health-related determinants and promoting health prevention measures.

Sedentary behavior (SB) is defined as any activities involving sitting or reclining at work, home, during travel, or with friends. This includes activities characterized by low energy expenditure (≤1.5 metabolic equivalents [METs]) [2], such as sitting at a desk, socializing, traveling by car, bus, or train, reading, playing cards, or watching television, but excludes sleeping [3]. A high level of SB has been defined as having at least 8 h of sedentary time per typical day [4]. 

Previous studies suggest that SB has little or no effect on physical activity, i.e., increased physical activity does not necessarily reduce sedentariness. A person may be physically active while still engaging in SB simultaneously [5,6,7,8]. Furthermore, substantial evidence has accumulated that shows that SB’s association with various adverse health outcomes is independent of health-related physical activity [9]. These outcomes include poor mental status (depression, dementia, cognitive impairment) [10,11,12], the main non-communicable diseases (hypertension, diabetes, metabolic syndrome, cardiovascular diseases) [5,13,14], falls [15], certain cancers [9], and increased risk of mortality [9,16,17,18]. Therefore, the two (SB and physical activity) need to be studied independently to understand their factors and their relationship with adverse health outcomes. 

Despite the growing body of evidence highlighting the adverse effects of SBs on health, there is a notable gap in the literature regarding the prevalence and correlates of SBs among the older Malaysian population. An analysis of nationally representative, population-based data collected in 2006 estimated the prevalence of a high level of SB (sitting time ≥ 6 h/day) among Malaysian adults aged 18 years and above at 32.7%, 44.7%, and 50.3% among older adults aged 61–70, 71–80, and over 80 years, respectively. However, this study did not thoroughly investigate the factors associated with SB, and failed to consider factors specific to this population [19]. Consequently, there remains a significant need for more in-depth research to fully understand the factors contributing to prolonged SB among the elderly in Malaysia. Several factors are associated with SB among the elderly, including sociodemographic characteristics (age, sex, marital status employment status, income, education) [20,21], lifestyle risks (tobacco and alcohol use), social activities, personal motorized transport, the presence of multimorbidity [22], functional factors (activities of daily living or instrumental activities of daily living [23]), verbal and vision abilities [24], quality of life [25] and environmental factors such as aspects of the indoor (type and size of house) and outdoor built environment (availability of public parks, walking pathways, low crime rate) [26]. Understanding these factors can help health authorities formulate preventive strategies and interventions to reduce SB and lower the risk of adverse health outcomes among the older population [27]. While factors associated with physical activity among older adults in Malaysia have been widely studied [28,29,30,31], there are sparse data on the correlates of SB among persons 65 years and older specifically [19]. Sedentary behavior is a growing public health concern globally, particularly among older adults, yet there is limited research on its determinants in culturally diverse settings such as Malaysia. Thus, in this study, we aimed to explore the associations of sociodemographic and lifestyle factors, mental health, nutritional status, social support, and functional limitations with SB among the older adults in Malaysia. This study offers a novel contribution by examining the unique sociocultural, environmental, and lifestyle factors that influence sedentary behavior among Malaysian older adults, a population experiencing rapid aging and urbanization. The study could be relevant to health policymakers seeking to understand how cultural and contextual factors shape sedentary behavior in non-Western settings, and can inform global efforts to promote active aging and reduce the health risks associated with sedentary lifestyles. 

## 2. Methodology

### 2.1. Study Design

We analyzed data from the National Health and Morbidity Survey (NHMS) 2018 on Elderly Health [32]. The NHMSs are nationwide population-based surveys conducted by the Ministry of Health Malaysia. NHMS 2018 was focused on elderly health and collected data for the purpose of informing the evaluation and revision of health priorities, program strategies, and activities, as well as for planning resource allocation for pre-elderly and elderly healthcare services. The survey consisted of 13 scopes/areas altogether, including socio-demography and living arrangement, public transport usage to access healthcare facilities, mental health status, functional status, physical activity, social support, nutritional status and dietary habits, quality of life, and health-related conditions including NCDs, urinary incontinence, oral health, visual and hearing impairment, and elder abuse. The present study was approved by the Malaysian Ministry of Health Medical Research Ethics Committee (NMRR-17-2655-39047).

### 2.2. Study Population

The study population in the NHMS 2018 comprised the non-institutional, middle-aged (age 50 to 59 years) and older adult (age 60 years and above) population in Malaysia. Non-institutional refers to not residing in institutional settings, such as hotels, hostels, hospitals, prisons, boarding houses, nursing homes, and similar institutions. However, our analysis in this study focused specifically on 3977 older adults aged 60 years and above (*n* = 3977).

### 2.3. Sampling Design

The sampling frame used in the NHMS 2018 was the National Household Sampling Frame issued by the Department of Statistics, Malaysia, which was updated in 2017. In the sampling frame, Malaysia is geographically divided into 83,000 Enumeration Blocks (EBs). Each EB contains, on average, between 80 and 120 LQs, and a population of between 500 and 600. The EBs are classified as either urban or rural based on population size and built-up area. Urban areas are gazetted areas that consist of adjoining built-up areas with a combined population of 10,000 or more. All other gazetted areas with populations less than 10,000 and non-gazetted areas are classified as rural areas. The NHMS 2018 employed a stratified cluster sampling design to ensure representativeness in the study population. The first stage was stratification by state/federal territory, followed by stratification by urban/rural classification. The sample size of each state was proportional to the states’ population size. The detailed sampling process has been published elsewhere [33] and will not be explained further here.

### 2.4. Data Extraction

#### 2.4.1. Dependent Variable

##### Sedentary Behaviour

Physical activity and SB were assessed using the Global Physical Activity Questionnaire (GPAQ) [3]. In the GPAQ, the item used for measuring SB was: “On a typical day, how much time do you usually spend sitting or reclining including time spent at work, at home, in leisure and during travel but not including time spent sleeping?” Participants were asked the amount of time spent sitting or reclining at work, home, during travel, or with friends, including on activities such as sitting at a desk, socializing, traveling by car, bus, or train, reading, playing cards, or watching television, but excluding sleeping. There is currently no universally accepted cut-off or definition of what constitutes a high level of SB. In this study, we used total of at least 8 sedentary hours on a typical day to define a high level of SB. This cutoff was selected based on previous research, which found that among the general population, sitting for ≥8 h per day was linked to an increased risk of premature mortality [4,18].

#### 2.4.2. Independent Variables

##### Sociodemographic Variables

Several sociodemographic characteristics of the participants selected for analysis were age (categorized as 60–64, 65–69, 70–74, 75–79, ≥80 years), gender (male or female), ethnicity (Malay, Chinese, Indian, Bumiputra Sarawak and Sabah, others), income (less than Ringgit Malaysia (MYR) 1000, MYR 1000–1999, MYR 2000 and above), marital status (single, married, separated/divorced, widow/widower), and employment status (employed, unemployed). Education level was categorized into no formal education, primary, secondary, and tertiary education (obtained formal post-secondary education including university, college, or technical training institutions). Private transport ownership was determined by the proxy question “mode of transportation used to access healthcare facilities” (for which the response options were private transport, public transport, and walking).

### 2.5. Lifestyle Factors and Comorbidity

Smoking habit was classified into three categories—current smoker, former smoker, and non-smoker. The physical activity level of respondents was assessed using the Global Physical Activity Questionnaire (GPAQ). According to GPAQ guidelines, an individual is considered “physically active” if they meet any of the following criteria: (i) engage in at least 30 min of moderate-intensity activity or walking per day, on at least 5 days in a typical week; (ii) participate in 20 min of vigorous-intensity activity per day on at least 3 days in a typical week; or (iii) accumulate at least 600 MET-minutes per week through any combination of walking and moderate- or vigorous-intensity activities across 5 days [3]. Respondents were also asked whether they have been medically diagnosed with diabetes, hypertension, hypercholesterolemia, or any form of cancer.

#### 2.5.1. Quality of Life

Quality of life was measured using the Summative Quality of Life (QoL) scale from the Control, Autonomy, Self-Realization, and Pleasure (CASP-19) questionnaire for assessing quality of life [34]. The CASP-19 comprises four domains—control (4 items), autonomy (5 items), pleasure (5 items), and self-realization (5 items), all scored on a 4-point Likert scale ranging from 0 to 3. The total score, ranging from 0 to 57, was obtained by summing all items, with higher scores indicating greater QoL satisfaction. Perceived poor quality of life (PPQoL) was classified for respondents with the QoL score ≤ 44 [35].

#### 2.5.2. Mental Health

Dementia was evaluated using the Identification and Intervention for Dementia in Elderly Africans (IDEA) Cognitive Screen, which has been validated for dementia screening in Malaysia [36]. IDEA scores of 10 or less were used to indicate possible dementia. The Geriatric Depression Scale (GDS-14), validated by Teh et al. (2004), was applied for depressive symptoms screening with a cut-off point of 6 or above to indicate clinically significant depression, and a score of 8 or above for major depression [37].

#### 2.5.3. Functional Capacity and Falls

Functional capacity refers to a person’s capacity to carry out physical tasks, including personal care, mobility, and maintaining independence both at home and in social environments. In this study, functional status was assessed using the Barthel Index of Activities of Daily Living, which measures activities of daily living (ADL), and the Lawton and Brody instrumental activities of daily living scale (IADL). Need for assistance due to inability to independently perform one or more ADLs indicates a functional limitation and need of supportive services. A total score of 20 on the Barthel Index indicates the absence of functional limitations, while scores below 20 indicate the presence of functional limitations [38]. Unlike the ADL, the IADL measures activities that are not necessarily performed daily but are required for independent living. The IADL score results were categorized into two groups—dependent (total score of 7 and below) and independent (total score of 8) [39]. Prevalence of falls was determined by history of falls in the twelve months prior to the date of interview.

#### 2.5.4. Visual and Hearing Disabilities

The questions used to assess vision and hearing disabilities were adapted from the Washington Group on Disability (WG) guidelines. The WG approach emphasizes functional limitations rather than impairments, making it suitable for international comparisons of prevalence rates [40]. Disability is an umbrella term that includes impairment, activity limitation, or participation restriction. Vision disability was classified based on respondents’ answers, with options of “no vision disability” or “vision disability”. Similarly, hearing disability was determined by responses indicating either “no hearing disability” or “hearing disability” [41].

#### 2.5.5. Nutritional Status

To identify individuals who are malnourished or at risk for malnutrition, the modified Mini Nutritional Assessment Short Form (MNA-SF) [42] was utilized as a screening tool. This instrument has been validated for use among older adults in Malaysia [43]. SF scores of 12 to 14 indicate normal nutritional status; scores of 8 to 11 indicate at risk of malnutrition; and scores 7 and below signify malnutrition. Additionally, calf circumference measurements were also taken as an indicator of malnutrition risk, as recommended by the WHO. The local cut-off point for calf circumference was used to determine the final MNA-SF score. Calf circumference values less than the cut-off of 30.1 cm for males or 27.3 cm for females were scored as “0”, whereas when exceeding 30.1 cm for males or 27.3 cm for females, they were scored as “3” [44].

#### 2.5.6. Anthropometric Indices

Body weight and height measurements were utilized to compute body mass index (BMI). BMI was calculated as the ratio of weight in kilograms to the square of height in meters (kg/m^2^) and categorized as follows: underweight (<18.5 kg/m^2^), normal weight (18.5–24.9 kg/m^2^), overweight (25.0–29.9 kg/m^2^), and obese (≥30.0 kg/m^2^). Waist circumference (WC) measurements were used to identify abdominal obesity, which was defined as waist circumference greater than 102 cm for men and greater than 88 cm for women [45]. 

#### 2.5.7. Social Support and Networking

The instrument used to measure social support in this module was the 11-item Duke Social Support Index (DSSI) [46]. The reliability and validity of the DSSI for the Malaysian population has been established in a previous study [47]. The 11-item DSSI comprises two sub-scales: the first measures the size and structure of the respondents’ social network (Social Interaction) and consists of four items, while the second is a seven-item subscale that measures perceived satisfaction with the behavioral or emotional support obtained from this network (Subjective Support) [47,48]. Social support is calculated as the sum of scores for items 1 to 11, with higher scores indicating greater social support. Established cut-off points categorize scores into low to fair (≤26), high (27–29), and very high (30–33) levels of social support, as published by Strodl et al. [49] for the Australian population, and these were used in this study to determine the prevalence of individuals with poor social support.

### 2.6. Statistical Analysis

Weighting factors were applied to each individual data point to adjust for non-response and unequal selection probability due to the complex sample design, to ensure generalizability to the older adult population of Malaysia. The dependent variable in the analysis was sedentary (Yes/No). The independent variables examined were sociodemographic factors (gender, age, ethnicity, marital status, education level, employment status, and monthly individual income), health risk factors (smoking status, physical activity, nutritional status), mental status (depression, probable dementia), functional limitations (ADL, IADL), history of falls, vision and hearing disabilities, mode of transportation, social support, and quality of life. Descriptive statistics (frequencies and percentages) were used to summarize these variables. Initially, Pearson’s chi-square tests were conducted to determine associations between these independent variables and a high level of SB (sitting for ≥8 h per day). This was followed by complex samples multiple logistic regression analysis to determine the association between these independent factors and the odds of engaging in sedentary activities. The final model fit was assessed using the receiver operating characteristic curve and classification table. No significant two-way interactions or multicollinearity were found between the variables. All statistical analyses were carried out at a 95% significance level using IBM SPSS Statistics for Windows, version 28 (IBM corp, Armonk, NY, USA).

## 3. Results

The total number of study respondents aged 60 years or above was 3977. Table 1 summarizes the characteristics of the respondents across various demographic, socio-economic, health, and lifestyle variables. The age distribution of respondents revealed that the majority were aged less than 70 years old (66.5%). The gender distribution was relatively balanced, with females comprising 51.1% of the respondents. Most of the respondents were Malay (57.7%), a significant majority of respondents were married (67.9%), and a large proportion of respondents received primary education (43.6%). In terms of employment, 75.7% of respondents were employed and 58.2% earned less than MYR 1000 per month.

The prevalence of prolonged SB was high, with 23.2% or almost a fourth of respondents classified as sedentary. A large majority of respondents were non-smokers (86.7%) and physically active (70.2%). Anthropometrically, 54.6% were overweight or obese (based on BMI), and 36.4% had abdominal obesity. Chronic diseases were prevalent among respondents, with 51.1% having hypertension, 41.8% having hypercholesterolemia, 27.7% having diabetes mellitus, and 1.6% having a cancer diagnosis. Mental health indicators showed depression in 11.2%, and probable dementia in 8.5% of the respondents. Most respondents had never experienced falls (85.9%), and had no hearing or vision disabilities (hearing disabilities 6.4%, vision disabilities 4.5%). Regarding activities of daily living (ADL), 83.0% of respondents were independent, while 17.0% required assistance. For instrumental activities of daily living (IADL), 42.9% of the respondents were unable to live independently. Nutritional status showed that 23.5% were at risk of malnutrition and 7.3% were malnourished. Most respondents did not live alone (93.7%) and owned a transportation vehicle (95.1%). Approximately 30.8% of respondents reported receiving inadequate social support, and 28.6% perceived their quality of life to be poor.

Univariate analysis with the Pearson chi-square test showed significant associations between several factors and a high level of SB among older adults in Malaysia (Table 2). Age was significantly associated with SB (*p* < 0.001), with older age groups showing higher percentages of SB. Occupational status was also a significant factor (*p* < 0.001), with unemployed individuals being more sedentary. Income level was significantly related *(p* = 0.004); those who earned less than MYR 1000 a month were more sedentary. Physical activity was strongly associated with SB (*p* < 0.001), with inactive individuals showing higher SB. Activities of daily living (ADL) status (*p* < 0.001) and instrumental activities of daily living (IADL) status (*p* = 0.002) were significantly related, with those dependent on others for their daily activities being more likely to be sedentary. Additionally, older persons who were malnourished showed a significant association with being sedentary (*p* = 0.005), while those who had their own transportation (*p* = 0.012) and whose perceived quality of life (*p* = 0.013) were good were less likely to engage in sedentary activities. A previous history of falls was significantly associated with more SB (*p* = 0.028).

The final sample size included in the multiple regression model after listwise deletion comprised 3455 respondents. The multiple regression model of the above factors, used to adjust for confounding, showed that several factors remained significant predictors of a high level of SB (Table 3). Older age groups exhibited higher odds of SB compared to those aged 60–64, with the AORs ranging from 1.58 to 2.76 (*p* < 0.001 to *p* = 0.001). Unemployment was also significantly associated with increased SB (AOR = 1.32, *p* = 0.02). Additionally, individuals with an income of MYR 1000–1999 had lower odds of SB compared to those with an income of ≥MYR 2000 (AOR = 0.64, *p* = 0.022). Bumiputra Sabah and Sarawak ethnicity showed a higher likelihood of SB compared to Malays (AOR = 2.48, *p* = 0.007). Those at risk of malnutrition had lower odds of being sedentary (AOR = 0.68, *p* = 0.031). Other factors, such as sex, marital status, education level, smoking status, physical activity, BMI status, abdominal obesity, chronic illness, depression, probable dementia, history of falls, disabilities, ADL and IADL status, living alone, transportation mode, social support, and perceived quality of life, were not significantly associated with SB.

## 4. Discussion

Our study found that the prevalence of SB among the older adult population in Malaysia, defined as time spent sitting or reclining for more than 8 h per day, was notably high, at 23.2%. The prevalence of SB increased with age; 17.1% in those aged 60–64 years, 22.8% in those aged 65–69 years, 25.5% in those aged 70–74 years, 33.2% in those aged 75–79 years, and 38.5% in those aged 80 years and above. In comparison, data from a previous nationwide survey of Malaysian adults aged 18 years and above in 2006 reported even higher prevalence rates of 32.7%, 44.7%, and 50.3% among older individuals aged 61–70, 71–80, and over 80 years old, respectively, but with SB defined as sitting or reclining for more than 6 h per day [19]. A comparative study involving nationally representative samples of adults aged 50 years and older from six countries, using data from the World Health Organization’s (WHO) longitudinal Study on Global Ageing and Adult Health (SAGE) between 2007 and 2010, reported widely varying SB prevalence rates (sitting or reclining ≥4 h/day): 45% in China, 58% in Russia, 43% in Ghana, 38% in India, 37% in South Africa, and 21% in Mexico [20]. Baseline data from a multicenter clinical trial in five European countries, which involved 2157 community-dwelling healthy older adults aged 70 and older, revealed that 37.1% of the participants spent ≥5 h/day with SB. The highest prevalence was reported in France (41.3%), followed by Austria (38.2%), Switzerland (37.8%), Germany (34.1%) and Portugal (33.6%) [8]. However, direct comparisons between our study and other studies may not be feasible due to large variations in study methodologies, such as study design, the definition of older adult, cut-offs for defining SB, and measurement tools used (objective measurement versus self-report method) for assessing SB.

This study provides valuable insights into the factors associated with SBs among older persons in Malaysia. Several significant associations were identified, suggesting a complex interplay of the effects of demographic, socioeconomic and health-related factors on sedentary behavior in this population. One of the key findings of our study was the significant dose–response relationship between age and SB. The positive relationship between age and sedentary behavior corroborates the findings of previous studies [19,20,21,50]. The adjusted odds ratios (AORs) ranged from 1.58 to 2.76, indicating that as age increases, so does the likelihood of being sedentary. This may be attributed to decreased physical capability, mobility issues, and increased health problems that limit activity levels. Data from the SAGE survey show that older age groups (≥65 years) with chronic conditions such as arthritis, chronic back pain, hearing problems, visual impairment and physical multimorbidity were more likely to engage in high levels of SB compared to those in the middle age group (50–64 years) [22]. A nationwide survey of individuals over 65 years old in Korea revealed that the oldest-aged (≥75 years) spent significantly more time sitting compared to the youngest old (65–74 years), likely due to the increased mobility impairments that often accompany the aging process [25]. The strong association underscores the need for targeted interventions aimed at reducing SB in the oldest segments of the older population, who may be at greater risk of associated chronic illness and health related complications. Suitable interventions for the very old may include increasing mobility around at home, performing light household chores, light gardening work such as watering plants, and taking care of pets.

Our study revealed that indigenous Sabah and Sarawak ethnicities had significantly higher odds of SB compared to Malays. However, the 2006 national health and morbidity survey reported that Malaysian adults of Chinese descent were more likely to engage in sedentary activities compared to other ethnicities [19]. A more recent study in 2019, among a multiethnic sample of Singaporeans aged 18 and above, showed higher odds of SB among the Chinese, although no statistically significant difference was found [51]. Cultural, socioeconomic and environmental changes over the past decade may account for these inconsistencies. For instance, changes in rural–urban living conditions, types of housing (landed or high-rise building), and the availability and accessibility to recreational and sports facilities [26], as well as socio-cultural and lifestyle practices [27], may vary significantly across ethnic groups, impacting physical activity levels and sedentariness. An in-depth qualitative study of the sociocultural differences among various ethnic groups and their influence on physical behaviors, particularly among older adults, is needed, as understanding these cultural nuances is crucial for developing culturally sensitive interventions aimed at reducing SB across the different ethnic groups.

Unemployment was another significant factor associated with sedentariness, with unemployed older persons exhibiting higher odds of being sedentary in the present study. This aligns with findings from the SAGE study conducted in China, India, South Africa and Ghana, where multivariate analysis also showed that older persons who were not working or were retired had higher odds (OR range: 1.5–2.0) of spending more than 4 h daily engaged in sedentary behavior compared to those who were employed [20]. Similar, a study involving 8273 community-dwelling older adults aged ≥65 years who participated in the Korean National Health and Nutrition Examination Survey reported that unemployed Korean older individuals were more likely to engage in sitting or reclining for more than 7 h per day compared to their employed counterparts [50]. These findings may reflect the lack of structured daily activities and social engagement often provided by employment. Several strategies could be developed to reduce SB, particularly targeting retired or unemployed older adults. These could include promoting an active lifestyle by engagement in social activities that are aligned with the interests and abilities of older adults. Examples include participating in volunteer work within the community, joining hobby groups such as hiking, camping, or birdwatching, and taking music, dance, or yoga classes [52]. Additionally, amending current labor laws to extend the minimum retirement age from 60 to 65, or allowing optional retirement at 60 for those not willing to work up to the age of 65, could also help reduce SB. In contrast to our findings, a cross-sectional study among a group of 397 older individuals who visited a polyclinic in Singapore found that SB was significantly associated with employment, with those who were employed having twice the odds of engaging in ≥8 h of SB daily [53]. Jobs that require prolonged sitting, such as managerial or administrative work, which involve the use of computers, making telephone calls, or paperwork are typically sedentary in nature. Similarly, occupations like driving public utility vehicles may contribute to higher levels of SB among employed persons. However, the study was limited by a small sample size and was confined to a single polyclinic.

Previous studies have consistently shown that older people from higher-income households are more likely to be sedentary [20]. Similar findings of less sedentariness among the lower-income group compared to the high-income group were also reported from the 2003–2006 NHANES among adult Americans [54]. In our study, the odds of engaging in SB among older persons with moderate incomes (MYR 1000–1999) were lower compared to those with higher incomes (≥MYR 2000). Lower-income older persons, often facing financial limitations, may be less able to afford sedentary lifestyles. In contrast, higher-income individuals may have greater access to amenities and conveniences that facilitate a more sedentary lifestyle. For example, older persons with lower incomes are more likely to use public transportation, such as buses or light rail, for travel, whereas those with higher incomes are more likely to use personal vehicles. Using public transport typically involves more walking and physical exertion, as lower-income older individuals often reside in less connected areas where bus stops or commuter stations are some distance from their homes [55]. Additionally, older persons with moderate incomes may live in less accessible areas, such as neighborhoods with fewer elevators, more stairs, and limited access to nearby convenience stores, restaurants or supermarkets for grocery shopping and dining, which require more physical movement. In contrast, higher-income older persons are more likely to live in more comfortable and accessible environments that require less physical effort.

Prolonged SB indirectly impacts the ability of older adults to perform activities of daily living, with this effect being mediated by nutritional risk [23]. Malnutrition or the risk of malnutrition increases the likelihood of developing sarcopenia [56], a condition characterized by low muscle mass, low muscle strength, and poor physical performance [57]. This may explain the detrimental effects of malnutrition on both SB and physical function. Interestingly, our study found that individuals at risk of malnutrition had lower odds of being sedentary. This counterintuitive finding suggests that those at nutritional risk may still be maintaining some level of moderate and vigorous physical activity, possibly out of necessity or in an attempt to improve their health status. The causal relationship between malnutrition and sedentary behavior cannot be definitively determined in this study. The observation that malnourished older adults are less likely to engage in sedentary behavior may stem from the fact that individuals who spend less time in sedentary activities are likely expending more energy. This increased energy expenditure, coupled with insufficient dietary intake, could contribute to a state of malnutrition. It also highlights the importance of comprehensive assessments in the older adult population, where nutritional status, as well as sedentary and physical activity, should be considered together to help better understand their health behaviors and outcomes [20,58]. Further studies should be undertaken on the complex interplay between energy–protein balance, sedentary level, and nutritional status in older adults.

There have been mixed findings regarding the relationships of factors such as sex, marital status, education level, lifestyle, physical function, mental health, and chronic illness with SB [20,59,60]. However, in the present study, these variables were not significantly associated with SB. Differences in study design, the sociodemographic characteristics of the study populations, and measurement methods may partly explain the inconsistent findings across studies. Furthermore, socio-cultural and environmental factors could mediate the relationship between these variables and SB.

One significant limitation of this study is its inherent inability to establish cause-and-effect relationships between the factors and SB due to the cross-sectional design. Additionally, reliance on self-reported data for time spent in sedentary activities may introduce recall bias, and objective measurements of SB using accelerometers would have provided more accurate assessments. Another limitation is that, if the respondent had a cognitive impairment or communication problems such as post-stroke effects, speech disabilities, or a language barrier, a proxy interview was conducted, whereby the proxy was a family member who knew the respondent best; thus, there may be proxy-reporting bias. Future research in this area should utilize longitudinal and qualitative study designs—the former, in order to better understand the temporal relationships between these factors and SB, and the latter, so as to obtain deeper insight into the cultural and socioeconomic contexts that contribute to the observed ethnic differences in SB. Despite the limitations mentioned, a key strength of this study is that the data were derived from a large, nationally representative and comprehensive population-based survey, which allowed us to conduct an in-depth examination of a wide range of potential predictors of SB, including sociodemographic characteristics, health risk factors, mental status, functional limitations, falls, mode of transportation, vision and hearing disabilities, social support, and quality of life.

## 5. Conclusions

These findings have important implications for public health initiatives aimed at reducing SB among older persons in Malaysia. Targeted interventions should focus on the oldest age groups and unemployed individuals, and policy-makers need to also consider ethnicity and cultural diversity when developing interventions. Additionally, efforts to improve nutritional status among the older persons should be integrated with programs to encourage physical activity, particularly in those who may not yet be highly sedentary. In conclusion, this study highlights the multifaceted nature of SB among the older adult population in Malaysia, pointing to specific demographic and socioeconomic factors that can inform targeted public health interventions such as community-based physical activity programs, awareness campaign, hourly movement reminders, and supportive environments. Addressing these factors is crucial for reducing SB and promoting healthier aging in this population.

## Figures and Tables

**Table 1 healthcare-13-00160-t001:** Respondents’ characteristics.

Variables	Estimated Population	*n*	% (95% CI)
Age group			
60–64	1,233,952	1442	38.2 (35.3, 41.2)
65–69	914,327	1121	28.3 (26.4, 30.3)
70–74	544,986	675	16.9 (15.3, 18.5)
75–79	293,874	429	9.1 (7.8, 10.6)
80+	243,202	310	7.5 (6.3, 8.9)
Sex			
Male	1,580,226	1872	48.9 (47.2, 50.6)
Female	1,650,114	2105	51.1 949.4, 52.8)
Ethnicity			
Malay	1,863,766	2591	57.7 (48.7, 66.2)
Chinese	855,542	710	26.5 (19.8, 34.5)
Indians	209,635	126	6.5 (4.1, 10.2)
Bumiputra Sabah and Sarawak	24,2023	436	7.5 (4.3, 12.8)
Others	59,365	114	1.8 (1.0, 3.5)
Marital status			
Single	98,087	87	3.0 (2.3, 4.0)
Married	2193,327	2623	67.9 (65.2, 70.5)
Separated or divorcee	57,859	64	1.8 (1.1, 2.8)
Widow or widower	879,595	1200	27.2 (24.5, 30.1)
Education level			
No formal education	469,794	806	14.5 (12.5, 16.9)
Primary	1,408,624	1939	43.6 (39.4, 47.9)
Secondary	1,040,544	967	32.2 (28.8, 35.8)
Tertiary	311,378	265	9.6 (7.4, 12.5)
Occupational status			
Unemployed	784,812	1050	24.3 (22.3, 26.4)
Employed	2,445,528	2927	75.7 (73.6, 77.7)
Income (MYR)			
<1000	1,851,033	2519	58.2 (54.5, 61.9)
1000–1999	682,569	845	21.5 (19.1, 24.1)
≥2000	645,096	567	20.3 (17.1, 23.9)
Smoking			
No	2,794,286	3346	86.7 (84.9, 88.3)
Yes	430,134	622	13.3 (11.7, 15.1)
Physical activity			
Active	2,263,127	2671	70.2 (66.9, 73.2)
Inactive	962,291	1,298	29.8 (26.8, 33.1)
Sedentary behaviours			
No	2,464,120	2999	76.8 (70.0, 82.4)
Yes	745,306	959	23.2 (17.6, 30.0)
BMI status			
Underweight	154,999	221	5.2 (4.2, 6.5)
Normal	1,197,044	1525	40.2 (37.7, 42.7)
Overweight	1,100,775	1292	37.0 (35.0, 39.0)
Obesity	525,242	610	17.6 (15.8, 19.6)
Abdominal obesity			
No	1,902,100	2401	63.6 (61.2, 66.0)
Yes	1,087,328	1275	36.4 (34.0, 38.8)
Chronic diseases (presence)			
Diabetes mellitus	891,213	1018	27.7 (25.5, 30.0)
Hypertension	1,645,628	2027	51.1 (48.9, 53.3)
Hypercholesterolemia	1,347,075	1576	41.8 (39.3, 44.4)
Cancer diagnosis	52,497	51	1.6 (1.1, 2.4)
Depression			
No	2,736,401	3,287	88.8 (86.6, 90.6)
Yes	346,126	485	11.2 (9.4, 13.4)
Probable dementia			
No	2,818,640	3366	91.5 (89.8, 93.0)
Yes	260,345	408	8.5 (7.0, 10.2)
Fall			
No	277,1494	3409	85.9 (84.2, 87.5)
Yes	453,675	560	14.1 (12.5, 15.8)
Presence of disability			
Vision disability	145,726	214	4.5 (3.4, 5.9)
Hearing disability	207,613	235	6.4 (3.4, 5.9)
ADL status			
Absent	2,674,188	3283	83.0 (80.8, 85.0)
Present	547,881	683	17.0 (15.0, 19.2)
IADL status			
Independent	1,840,829	2042	57.1 (54.0, 60.1)
Dependent	1,384,111	1925	42.9 (39.9, 46.0)
Nutritional status			
Not malnourished	2,233,784	2558	69.2 (66.1, 72.0)
At risk of malnutrition	760,140	1080	23.5 (21.2, 26.0)
Malnourished	236,416	339	7.3 (6.0, 8.9)
Not malnourished	2,233,784	2558	69.2 (66.1, 72.0)
Living alone			
No	3,027,143	3682	93.7 (92.5, 94.7)
Yes	203,198	295	6.3 (5.3, 7.5)
Transportation			
Public	131,735	208	4.1 (2.6, 6.3)
Own	3,069,040	3727	95.1 (92.8, 96.6)
walking	27,864	37	0.9 (0.4, 1.7)
Poor social support			
No	2,227,758	2698	69.2 (65.5, 72.8)
Yes	989,806	1261	30.8 (27.2, 34.5)
Perceived poor quality of life			
No	2,171,526	2467	71.4 (67.5, 75.0)
Yes	868,670	1283	28.6 (25.0, 32.5)

**Table 2 healthcare-13-00160-t002:** Associations of sociodemographic and lifestyle factors, mental health, nutritional status, social support, and functional limitations with sedentary behaviors among older adults in Malaysia.

Variables	Sedentary Behavior				
	No		Yes		** p*-Value
	*n*	% (95% CI)	n	% (95% CI)	
Age group					
60–64	1166	82.9 (76.4, 87.9)	270	17.1 (12.1, 23.6)	<0.001
65–69	863	77.2 (69.7, 83.2)	253	22.8 (16.8, 30.3)	
70–74	508	74.5 (65.7, 81.6)	166	25.5 (18.4, 34.3)	
75–79	283	66.9 (56.4, 75.7)	143	33.2 (24.3, 43.6)	
80+	179	61.5 (50.4, 71.6)	127	38.5 (28.4, 49.6)	
Sex					
Male	1414	77.1 (70.2, 82.9)	453	22.9 (17.1, 29.8)	0.676
Female	1585	76.4 (69.4, 82.3)	506	23.6 (17.7, 30.6)	
Ethnicity					
Malay	2029	78.8 (70.8, 85.2)	545	21.2 (14.8, 29.2)	0.240
Chinese	563	77.9 (65.5, 86.8)	145	22.1 (13.2, 34.5)	
Indians	91	73.7 (50.0, 88.7)	35	26.3 (11.3, 50.0)	
Bumiputra Sabah and Sarawak	250	61.9 (50.0, 72.5)	186	38.1 (27.5, 50.0)	
Others	66	67.4 (50.4, 80.7)	48	32.6 (19.3, 49.6)	
Marital status					
Single	67	81.0 (67.2, 89.9)	19	19.0 (10.1, 32.8)	0.229
Married	2024	77.6 (70.6, 83.4)	587	22.4 (16.6, 29.4)	
Separated or divorcee	48	81.7 (65.2, 91.4)	16	18.3 (8.6, 34.8)	
Widow or widower	857	73.8 (66.4, 80.1)	337	26.2 (19.9, 33.6)	
Education level					
No formal education	538	68.4 (61.0, 75.0)	264	31.6 (25.0, 39.0)	0.07
Primary	1501	77.8 (70.7, 83.6)	426	22.2 (16.4, 29.3)	
Secondary	747	78.2 (69.3, 85.1)	218	21.8 (14.9, 30.7)	
Tertiary	213	79.9 (68.5, 87.8)	51	20.1 (12.2, 31.5)	
Occupational status					
Unemployed	2133	74.8 (67.6, 80.8)	778	25.2 (19.2, 32.4)	<0.001
Employed	866	83.0 (76.7, 87.9)	181	17.0 (12.1, 23.3)	
Income (MYR)					
<1000	1803	73.4 (66.4, 79.4)	701	26.6 (20.6, 33.6)	0.004
1000-1999	711	83.5 (76.4, 88.8)	133	16.5 (11.2, 23.6)	
≥2000	454	79.5 (69.4, 86.8)	113	20.5 (13.2, 30.6)	
Smoking					
No	2526	76.8 (70.0, 82.5)	807	23.2 (17.5, 30.0)	0.840
Yes	470	76.3 (68.1, 82.9)	152	23.7 (17.1, 31.9)	
Physical activity					
Active	2132	80.5 (73.3, 86.1)	536	19.5 (13.9, 26.7)	<0.001
Inactive	866	68.0 (59.4, 75.5)	423	32.0 (24.5, 40.6)	
BMI status					
Underweight	157	73.4 (61.8, 82.5)	64	26.6 (17.5, 38.2)	0.429
Normal	1184	77.8 (70.9, 83.4)	339	22.2 (16.6, 29.1)	
Overweight	1021	79.6 (71.8, 85.6)	270	20.4 (14.4, 28.2)	
Obesity	471	78.5 (69.8, 85.2)	136	21.5 (14.8, 30.2)	
Abdominal obesity					
No	1877	78.1 (72.1, 84.1)	521	21.3 (15.9, 27.9)	0.394
Yes	965	77.2 (68.8, 83.9)	306	22.8 (16.1, 31.2))	
Chronic diseases (presence)					
Diabetes mellitus	762	76.4 (67.8, 83.3)	251	23.6 (16.7, 32.2)	0.811
Hypertension	1521	76.6 (69.2, 82.6)	501	23.4 (17.4, 30.8)	0.766
Hypercholesterolemia	1160	75.5 (67.8, 81.9)	411	24.5 (18.1, 32.2)	0.165
Cancer diagnosis	36	68.3 (50.1, 82.2)	15	31.7 (17.8, 49.9)	0.185
Depression					
No	2560	78.2 (71.2, 83.9)	719	21.8 (16.1, 28.8)	0.092
Yes	335	72.6 (63.3, 80.2)	148	27.4 (19.8, 36.7)	
Probable dementia					
No	26.9	77.9 (70.8, 83.7)	751	22.1 (16.3, 29.2)	0.439
Yes	286	75.1 (66.4, 82.1)	114	24.9 (17.9, 33.6)	
History of falls					
No	2595	77.7 (71.1, 83.2)	800	22.3 (16.8, 28.9)	0.028
Yes	402	71.1 (61.4, 79.2)	158	28.9 (20.8, 38.6)	
Presence of disabilities					
Vision diabilities	150	71.6 (58.3, 82.0)	64	28.4 (18.0, 41.7)	0.310
Hearing disabilities	158	76.8 (66.0, 84.9)	71	23.2 (15.1, 34.0)	0.994
Presence of functional limitation (ADL)					
No	2570	79.0 (72.1, 84.5)	705	21.0 (15.5, 27.9)	<0.001
Yes	423	65.7 (56.5, 73.8)	252	34.3 (26.2, 43.5)	
Limitations in instrumental activities of daily living (IADL)					
No	1595	80.0 (72.9, 85.6)	442	20.0 (14.4, 27.1)	0.002
Yes	1400	72.5 (65.1, 78.8)	515	27.5 (21.2, 34.9)	
Nutritional status					
Not malnourished	1952	77.6 (70.3, 83.5)	604	22.4 (16.5, 29.7)	0.005
At risk of malnutrition	842	78.3 (70.1, 84.7)	233	21.7 (15.3, 29.9)	
Malnourished	205	63.9 (55.5, 71.5)	122	36.1 (28.5, 44.5)	
Living alone					
No	2788	76.9 (70.2, 82.5)	875	23.1 (17.5, 29.8)	0.601
Yes	211	75.1 (64.5, 83.3)	84	24.9 (16.7, 35.5)	
Transportation					
Public	116	64.1 (52.2, 74.5)	92	35.9 (25.5, 47.8)	0.012
Own transport	2858	77.4 (70.6, 83.0)	852	22.6 (17.0, 29.4)	
walking	22	67.5 (46.7, 83.1)	14	32.5 (16.9, 53.3)	
Poor social support					
No	2062	77.8 (71.0, 83.4)	631	15.4 (11.5, 20.3)	0.227
Yes	927	74.3 (65.6, 81.4)	326	25.7 (18.6, 34.4)	
Perceived poor quality of life					
No	1947	79.6 (72.4, 85.3)	516	20.4 (14.7, 27.6)	0.013
Yes	932	72.3 (64.1, 79.2)	344	27.7 (20.8, 35.9)	

IADL, instrumental activities of daily living. * Pearson Chi-square was performed.

**Table 3 healthcare-13-00160-t003:** Factors associated with sedentary behaviors among the older persons in Malaysia.

Independent Variables	AOR (95% CI)	*p*-Value
Age group		
60–64	reference	
65–69	1.58 (1.22, 2.06)	<0.001
70–74	1.78 (1.15, 2.75)	0.010
75–79	2.25 (1.39, 3.63)	0.001
80+	2.76 (1.49, 5.10)	0.001
Sex		
Female	reference	
Male	1.01 (0.69, 1.46)	0.97
Ethnicity		
Malay	reference	
Chinese	1.01 (0.45, 2.25)	0.982
Indians	1.84 (0.65, 5.19)	0.244
Bumiputra Sabah and Sarawak	2.48 (1.29, 4.76)	0.007
Others	1.37 (0.57, 3.31)	0.477
Marital status		
Single	reference	
Married	1.40 (0.74, 2.61)	0.293
Separated or divorcee	1.16 (0.39, 3.45)	0.782
Widow or widower	1.21 (0.61, 2.39)	0.577
Education level		
No formal education	reference	
Primary	0.85 (0.58, 1.22)	0.368
Secondary	1.11 (0.64, 1.92)	0.717
Tertiary	0.82 (0.39, 1.72)	0.597
Occupational status		
Unemployed	1.32 (1.05, 1.67)	0.02
Employed	reference	
Income (MYR)		
<1000	1.00 (0.68, 1.47)	0.993
1000–1999	0.64 (0.44, 0.94)	0.022
≥2000	reference	
Smoking		
Yes	1.33 (0.94, 1.88)	0.104
No	reference	
Physical activity		
Inactive	1.37 (0.93, 2.00)	0.107
Active	reference	
Body mass index		
Underweight	reference	
Normal	0.84 (0.53, 1.33)	0.464
Overweight	0.77 (0.47, 1.27)	0.302
Obesity	0.80 (0.46, 1.40)	0.435
Abdominal obesity		
Yes	1.18 (0.86, 1.63)	0.306
No	reference	
Presence of chronic diseases		
Diabetes mellitus	0.96 (0.74, 1.26)	0.770
Hypertension	0.87 (0.68, 1.11)	0.261
Hypercholesterolemia	1.25 (0.95, 1.63)	0.108
Cancer diagnosis	1.205 (0.53, 2.75)	0.654
Depression		
Yes	0.91 (0.63, 1.30)	0.594
No	reference	
Probable dementia		
Yes	0.66 (0.43, 1.01)	0.055
No	reference	
Falls		
Yes	1.14 (0.80, 1.62)	0.477
No	reference	
Presence of disability		
Vision disability	0.82 (0.41 (1.66)	0.576
Hearing disability	0.61 (0.35, 1.08)	0.086
Presence of functional limitation (ADL)		
Yes	1.20 (0.81, 1.78)	0.360
No	reference	
Limitations in instrumental activities of daily living (IADL)		
Yes	1.12 (0.84, 1.50)	0.427
No	reference	
Nutritional status		
At risk of malnutrition	0.68 (0.48, 0.96)	0.031
Malnutrition	0.83 (0.46, 1.51)	0.542
No malnutrition	reference	
Living alone		
Yes	0.99 (0.65, 1.50)	0.944
No	reference	
Transportation		
Public	0.71 (0.28, 1.83)	0.479
Own transport	0.56 (0.23, 1.40)	0.216
Walking	reference	
Poor social support		
Yes	1.01 (0.69, 1.49)	0.954
No	reference	
Perceived poor quality of life		
Yes	1.21 (0.83, 1.76)	0.319
No	reference	

ADL = activities of daily living; IADL = instrumental activities of daily living; AOR = adjusted odds ratio Complex samples logistic regression analysis was employed. The model fit was assessed using the receiver operating characteristic curve (Area under the curve = 0.672, *p* < 0.001) and percent of correct classification of 78.5%. No significant two-way interactions or multicollinearity were found between the variable (*p* > 0.05).

## Data Availability

Data are available upon reasonable request. The dataset used and analyzed in this study is available from the National Institutes of Health, Ministry of Health Malaysia upon reasonable request (https://nihdars.nih.gov.my/login, accessed on 12 March 2024) and with permission from the Director General of Health, Malaysia.

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
