# Peer review of "Sedentary Behaviour and Its Correlates Among Older Adults in Malaysia"

_healthcare, 2025, doi:10.3390/healthcare13020160_

Round 1

Reviewer 1 Report

Comments and Suggestions for Authors

The study investigated the associations between different sociodemographic, lifestyle, health and functional characteristics/variables and sedentary behaviour in a sample of older persons in Malaysia, using data extracted from the Malaysian National Health and Morbidity Survey. The topic is relevant, primarily under the public health perspective of the state of Malaysia, but also contributes to the international public health body of knowledge. The methods of data analysis are appropriate. The results are presented clearly and the conclusions are consistent with the evidence and presented arguments.

Please, consider substituting the term "elderly" with "older persons" or similar throughout the manuscript, since it is now considered as pejorative, or connoting discrimination (please see: Correct and Preferred Usage | AMA Manual of Style: A Guide for Authors and Editors | AMA Manual of Style | Oxford Academic).

Minor comment: page 2, line 91 - either "The survey contained 13 scopes" or "The survey consisted of 13 scopes"

Author Response

Comment 1: The study investigated the associations between different sociodemographic, lifestyle, health and functional characteristics/variables and sedentary behaviour in a sample of older persons in Malaysia, using data extracted from the Malaysian National Health and Morbidity Survey. The topic is relevant, primarily under the public health perspective of the state of Malaysia, but also contributes to the international public health body of knowledge. The methods of data analysis are appropriate. The results are presented clearly and the conclusions are consistent with the evidence and presented arguments.

Response 1: Thank you for the positive comment.

Comment 2: Please, consider substituting the term "elderly" with "older persons" or similar throughout the manuscript, since it is now considered as pejorative, or connoting discrimination (please see: Correct and Preferred Usage | AMA Manual of Style: A Guide for Authors and Editors | AMA Manual of Style | Oxford Academic).

Response 2: All instances of ‘elderly’ in the manuscript have been replaced with ‘older’, ‘older persons’, ‘older adults’, or ‘persons 60 years and older’.

Comment 3: Minor comment: page 2, line 91 - either "The survey contained 13 scopes" or "The survey consisted of 13 scopes"

Response 3: Thank you for pointing this out. We have revised the sentence to  ”The survey consisted of 13 scopes…"

Reviewer 2 Report

Comments and Suggestions for Authors

- There appear to be additional limitations in this study. Please consider and include these limitations.

- The participants in this study are older adults. Are all of them cognitively intact? Do any of them have cognitive issues such as dementia or mild cognitive impairment? Or was it impossible to confirm this?

- The finding that the likelihood of SB is lower in the middle-income group compared to the high-income group seems somewhat paradoxical.

- While the discussion interprets the results well, it could benefit from stronger comparisons with international studies. Additionally, please provide a more detailed explanation of the finding that individuals at nutritional risk are less likely to engage in SB.

- The necessity of tailored interventions for older adults is well emphasized, but including representative examples would help readers understand more easily.

Author Response

Comment 1: There appear to be additional limitations in this study. Please consider and include these limitations.

The participants in this study are older adults. Are all of them cognitively intact? Do any of them have cognitive issues such as dementia or mild cognitive impairment? Or was it impossible to confirm this?

Response 1:

We are unable to determine whether respondents had confirmed cognitive impairment /dementia, however, a total 400 respondents had probable dementia using the Identification and Intervention for Dementia in Elderly Africans (IDEA) screening instrument.

We have added the following limitation: “If the respondent had cognitive impairment or communication problems such as post-stroke, speech disabilities, or language barrier, proxy interview was conducted, whereby the proxy was a family member who knew the respondent best, thus there may be proxy-reporting bias.” Page 15 Line 439-442.

Comment 2: The finding that the likelihood of SB is lower in the middle-income group compared to the high-income group seems somewhat paradoxical.

Response 2: We contend that this finding is hardly paradoxical. Similar findings of less sedentariness among the lower-income group compared to the high-income group was also reported from the 2003-2006 NHANES among adult Americans [57]. We have offered several plausible explanation for this in the discussion, but another likely explanation that we may add is the higher income group have less frequent, but higher intensity physical activity and thus more sedentary time, as reported by Shuval et al.

Ref. 57. Shuval K, Li Q, Gabriel KP, Tchernis R. Income, physical activity, sedentary behavior, and the 'weekend warrior' among U.S. adults. Prev Med. 2017 Oct;103:91-97. 

Comment 3: While the discussion interprets the results well, it could benefit from stronger comparisons with international studies. Additionally, please provide a more detailed explanation of the finding that individuals at nutritional risk are less likely to engage in SB.

Response 3: We have cited the findings of other international studies on this topic (References number 20, 23, 58, 59, 60). And we have also added further explanation of this finding that individuals at nutritional risk are less likely to engage in SB (Page 15, line 421-431). “The causal relationship between malnutrition and sedentary behavior cannot be definitively determined in this study. The observation that malnourished older adults are less likely to engage in sedentary behavior may stem from the fact that individuals who spend less time in sedentary activities are likely expending more energy. This increased energy expenditure, coupled with insufficient dietary intake, could contribute to a state of malnutrition. Further studies should be undertaken on the complex interplay between energy-protein balance, sedentary level, and nutritional status in older adults.“

Comment 4: The necessity of tailored interventions for older adults is well emphasized, but including representative examples would help readers understand more easily.

Response 4:  We have added some examples: Suitable interventions for the very old may include increasing mobility around at home, performing light household chores, light gardening work such as watering plants, and taking care of pets (Page 14 line 349-351). Examples of interventions for unemployed older adults have been outlined on Page 18, line 378-385

Reviewer 3 Report

Comments and Suggestions for Authors

Abstract

Sedentary behavior should be briefly defined in the abstract

Introduction

The authors made a good illustration of the background concerning sedentary behavior in Malaysia, associated risk factors, and health consequences. However, the study is lacking novelty. Additionally, the study in the current format is not adding further knowledge that can be of interest to international audience and is mainly related to a local Malaysian context.  It would be appreciated if the authors could make a narration concerning why this study is important and how can it benefit other societies in international context.

Methodology

The study is based on secondary analysis of previously collected data. The authors made a good description of methods used to assess the data and referred to the primary studies accordingly.

Results

the first paragraph should declare the total sample size used in the final analysis and whether there are excluded cases.

table one: please explain what is meant by tertiary education.

Table two is not cited in the text. Please cite the table accordingly.

Discussion

the authors made a good narrative comparing the current findings to similar findings detected by other studies. However, one main limitation that might affect the public health implications of the current study is its reliance on statistical interpretation of the logistic regression analysis with limited public health impact considerations. For example, the authors claimed that targeting Malaysians whom are indigenous from Sabah and Sarawk is warranted based on the findings of their logistic regression. However, it can be noted that the OR associated we SB among people from this ethic group (2.48) is estimated based on the comparison to Malays as a reference. Furthermore, claiming that targeting Individuals from Sabah and Sarawak by interventions to reduce SB might not be optimum because it does not consider the small proportion of individuals from Sabah and Sarawak in the current sample (only 7.5%) and targeting these group as a recommendation for practical implications may reduce the public health impact that can occur when targeting individuals with SB among Malay and Chinese ethnicities (who contribute 84% of the sample).

Author Response

Comment 1: Abstract

Sedentary behavior should be briefly defined in the abstract.

Response 1: We have added a brief definition of sedentary behaviour in the abstract as follows “Sedentary behavior (SB), which are low-energy, wakeful activities performed in a sitting, reclining, or lying posture…” (Page 1, line 14)

Comment 2: Introduction

The authors made a good illustration of the background concerning sedentary behavior in Malaysia, associated risk factors, and health consequences. However, the study is lacking novelty. Additionally, the study in the current format is not adding further knowledge that can be of interest to international audience and is mainly related to a local Malaysian context.  It would be appreciated if the authors could make a narration concerning why this study is important and how can it benefit other societies in international context.

Response 2: Thank you for your suggestion. We have added some information on what new knowledge can be gleaned from this study, and its relevance to international readers.

“Sedentary behavior is a growing public health concern globally, particularly among older adults, yet there is limited research on its determinants in culturally diverse settings such as Malaysia. This study offers a novel contribution by examining the unique sociocultural, environmental, and lifestyle factors that influence sedentary behavior among Malaysian older adults, a population experiencing rapid aging and urbanization. The study could be relevant to health policymakers seeking to understand how cultural and contextual factors shape sedentary behavior in non-Western settings and can inform global efforts to promote active aging and reduce the health risks associated with sedentary lifestyles.”(Page 2, line 82-92).

Comment 3: Methodology

The study is based on secondary analysis of previously collected data. The authors made a good description of methods used to assess the data and referred to the primary studies accordingly.

Response 3: Thank you for the positive comment.

Comment 4: Results

the first paragraph should declare the total sample size used in the final analysis and whether there are excluded cases.

table one: please explain what is meant by tertiary education.

Response 4: We have provided the following information on sample size. “The total study respondents aged 60 years or above was 3,977” (Page 6, line 245). “The final sample size included in the multiple regression model after listwise deletion comprised of 3,455 repondents” (Page 11, line 292).

We have added the following definition of tertiary education in the Methods section.

“Education level was categorised into no formal education, primary, secondary, and tertiary education (Obtained formal post-secondary education including university, college, or technical training institutions)” (Page 3, line 142).

Comment 5: Table two is not cited in the text. Please cite the table accordingly.

Response 5: Reference to Table 2 has been added (Page 8, line 271).

Comment 6:

Discussion

the authors made a good narrative comparing the current findings to similar findings detected by other studies. However, one main limitation that might affect the public health implications of the current study is its reliance on statistical interpretation of the logistic regression analysis with limited public health impact considerations. For example, the authors claimed that targeting Malaysians whom are indigenous from Sabah and Sarawak is warranted based on the findings of their logistic regression. However, it can be noted that the OR associated we SB among people from this ethic group (2.48) is estimated based on the comparison to Malays as a reference. Furthermore, claiming that targeting Individuals from Sabah and Sarawak by interventions to reduce SB might not be optimum because it does not consider the small proportion of individuals from Sabah and Sarawak in the current sample (only 7.5%) and targeting these group as a recommendation for practical implications may reduce the public health impact that can occur when targeting individuals with SB among Malay and Chinese ethnicities (who contribute 84% of the sample).

Response 6: We appreciate and concur with the reviewer’s insightful comment. Though Sabah and Sarawak indigenous groups may have higher odds of SB, they indeed constitute a relatively small proportion of the population. Thank you for pointing out this error, we have thus revised our recommendation thus: “Targeted interventions should focus on the oldest age groups and unemployed individuals, and policy-makers need to also consider ethnicity and cultural diversity when developing interventions” on Page 16, Line 449-451.

Round 2

Reviewer 2 Report

Comments and Suggestions for Authors

The authors have adequately addressed the reviewers' comments through appropriate revisions and responses. They have acknowledged the limitations of the study and added additional points to address them. Other responses are logical, and I respect the authors' perspectives. Therefore, I recommend accepting the manuscript in its current form.

Reviewer 3 Report

Comments and Suggestions for Authors

The authors have addressed all my comments and have modified the manuscript accordingly.